# Job Burnout and Job Satisfaction among Healthcare Service Providers in a Daycare Center for Individuals with Autism Spectrum Disorders in Low-Resource Settings

**DOI:** 10.3390/brainsci13020251

**Published:** 2023-02-01

**Authors:** Sayyed Ali Samadi, Cemal A. Biçak, Nigar Osman, Barez Abdalla, Amir Abdullah

**Affiliations:** 1Institute of Nursing and Health Research, Ulster University, Belfast BT15 1ED, UK; 2Manager at New Breeze Autism Center (NBAC), Dihok 42001, Iraq; 3Independent Researcher, Erbil 44002, Iraq

**Keywords:** job burnout, job satisfaction, autism spectrum disorders, Kurdistan Region of Iraq, healthcare, low-resource settings

## Abstract

Job satisfaction and burnout are components of job morale. In general, and among healthcare provider personnel, these are psychological factors of the job and under the influence of different conditions and the organizational management of the healthcare systems. Both job burnout and job satisfaction among healthcare service providers have received scant attention in the literature, particularly in the healthcare systems of the Kurdistan Region of Iraq (KRI) as one low- or middle-income country (LMIC). The burnout rate and job satisfaction in a daycare center for children with autism spectrum disorders were reviewed and measured using a sample consisting of 34 employees from three different sections. The Maslach Burnout Inventory-Third Edition (MBI-3) and the Job Descriptive Index (JDI) were used. The relationships between the two scales and their consisting factors were examined using Pearson Correlation and Chi-square test to understand the correlation and levels of significant difference between the expected and the observed frequencies. There was a significant negative correlation between job burnout and satisfaction with the job and some significant correlations between the factors of the scales. Lower levels of emotional exhaustion and depersonalization factors of the burnout scale were statistically correlated. It was shown that the personnel were mainly satisfied with their jobs through their choices in the four parts of the job satisfaction scale. Further investigations are needed to understand different contributing factors to job satisfaction and burnout among healthcare providers in KRI. The current study might highlight the importance of understanding the healthcare providers’ perspectives on their careers.

## 1. Introduction

Autism Spectrum Disorder (ASD), as explained in the Fifth Edition of the Diagnostic and Statistical Manual of Mental Disorders (DSM-5), is categorized as a lifelong neurodevelopmental disorder. The presence of persistent deficits in social communication and social interaction across multiple contexts and the manifestation of restricted, repetitive patterns of behavior, interests, or activities are the core symptoms and dyad impairments in this diagnosis [1]. Based on the available report, the global prevalence rate is around 1 to 2% of children [2]. There are different prevalence rates for different countries. There is no exact prevalence rate in many low- and middle-income countries (LMICs). Hence, there is a shortage of studies about the prevalence of ASD in less-affluent countries, and most studies are investigated in high-income countries (HICs) [3]. Observing the KRI’s health system for individuals with ASD provides many first-hand lessons about available ASD service reform barriers [4]. There are different aspects of the impacts of this population on various groups, such as caregivers, service providers, and other factors that deserve to be understood. The wealth of data on family, friends, and caregivers have shown that they are significantly and negatively impacted by caregiving.

Nevertheless, healthcare service providers’ personnel have rarely been investigated to understand the different aspects of these demands [5]. This neglect is more evident in low- and middle-income countries (LMICs). The essential element of this shortage is the absence of data on theoretical or conceptual models of understanding the role that supports personnel’s perspectives about their jobs and their associated commitments. There are some studies regarding the impacts of service providing for a particular group of individuals with developmental disabilities (DD), such as ASD [6] or intellectual disabilities (ID) [5], as well as specific approaches to service providing, such as behavioral analysts [7,8,9,10,11,12] or DD in general [13]. It is found that interacting with clients who may show various problems leads to psychological helplessness and mental exhaustion [14]. Several studies showed that the impacts of long-term service providing on healthcare service providers’ personnel could be tracked through aspects such as job burnout and job satisfaction. Ultimately, the mentioned job qualities can harm the quality of care provided to individuals with ASD. Job qualities are important because they are related to job morale, and in the field of healthcare, job morale is widely seen as an essential detector of the quality of care. Job morale is not a single entity and comprises predictor variables, such as job motivation, job satisfaction, and job burnout [15].

There are data on issues labeled as negative provider behaviors among the healthcare providers in some LMICs that signified dissatisfaction of the service users and increased the provocation of verbal harassment, physical attack, and death threats towards providers [16]. The inability to identify and address contributing factors to negative provider behavior could have appalling consequences for service users and providers. Therefore, finding feasible and sustainable means of predicting and preventing negative provider behaviors warrants greater satisfaction for users and providers of different services for individuals with developmental disabilities. Finding the predicting factors contributing to boosting the overall satisfaction of healthcare providers in the Kurdistan Region of Iraq (KRI) seems crucial for the healthcare organization’s policymakers and managers. It provides promising, feasible, and sustainable recommendations to address the systemic challenge of lower levels of primary motivation among healthcare services providers, which has also been reported in other studies in other LMICs [17].

It is found that higher levels of employee burnout and dissatisfaction are associated with poorer service user outcomes and are considered to be “contagion” and negatively impact the workforce [18]. The two factors of job burnout and job satisfaction are considered for investigation in the current study. There were three main aims for this study, and an analysis was done to attain the following objectives among the healthcare providers in a daycare center for children with ASD to understand the following:There is a significant difference between the level of job burnout among rehabilitation and training personnel of healthcare providers compared to the services and administration personnel.There is a significant difference between the level of job satisfaction among rehabilitation and training personnel of healthcare providers compared to the services and administration personnel.There are significant correlations between factors of job burnout and satisfaction with the job.

### 1.1. Job Burnout

Morse et al. [19] found that burnout is common among healthcare providers. It is estimated that up to 67% of mental healthcare service providers experience higher degrees of burnout. Poghosyan et al. [20] findings indicate that inequity between work-related demands and personal resources leads to job burnout manifested through the three dimensions of emotional exhaustion, depersonalization, and reduced personal accomplishment. Burnout is a term used to describe feelings of emotional, physical, and mental fatigue induced by different sources of stress. Burnout is a syndrome that comprises three main components with signs such as tiredness; fewer emotional resources (first component: emotional exhaustion) and development of negative, cynical attitudes; impersonal treatment of clients (second component: depersonalization); and thoughts of uselessness and insufficiency (third component: feelings of a lack of personal accomplishment) [21,22]. A recent review indicated a 21–67% burnout rate among mental health providers, and it is concluded that burnout is a common factor among mental healthcare service providers [19].

### 1.2. Job Satisfaction

Based on Gebert and von Rosenstiel’s [23] suggestion, job satisfaction results from comparing the desired to the actual work situation. A well-accepted definition of job satisfaction is presented by Spector [24], who says that job satisfaction is “the extent to which people like (satisfaction) or dislike (dissatisfaction) their jobs”. Job satisfaction is a predictor of psychological well-being [25] and an indicator of strain. Moreover, it is often conceptualized as a health-related outcome [26,27]. Healthcare providers’ job satisfaction is influenced by various psychological, individual, and organizational factors, such as personality, work type, payment, assigned duties, administration, and leadership styles [28]. Low job satisfaction might cause work stress and burnout [29] and might be considered factors contributing to healthcare turnover rates.

Although burnout and lack of satisfaction with the job occur throughout different job types, available literature indicates a higher rate for mental health providers due to the other demands associated with this group of healthcare service providers [30]. Globally, efforts have begun to explore contributing factors and predictors of job burnout and job satisfaction to assist in designing effective interventions to control burnout factors and to understand job satisfaction. The efforts are made because having the desire to leave the job, feelings of burnout, and not being satisfied with the job inhibits the ability of optimal job performance fulfillment and consequently negatively impacts service provision. Morse et al. [19] reported negative impacts of burnout on intervention outcomes, particularly for personnel who report severe burnout. Similar findings were reported about clients who receive services from a team with a high level of exhaustion due to job burnout; they generally feel less satisfied with the delivered services [31].

### 1.3. The Present Study

There is a shortage of data about healthcare personnel job burnout and job satisfaction in most Middle Eastern countries, and consequently in the Kurdistan Region of Iraq (KRI), mainly for the healthcare services-providing groups for individuals with ASD. Still, burnout is a more well-known factor to study in this geographical area. Some findings of job burnout among healthcare service providers [32] and healthcare service providers for individuals with DD [6] are available. Chemali and colleagues [33], in a review of 54 published papers, investigated this factor among physicians and reported a high burnout level. Green et al. [30] reported a similar shortage in more affluent and developed countries. They said that regardless of the increased risk for factors such as job burnout among care workers in the mental health sector, presently, there is a shortage of studies comparing experiences and contributing factors such as burnout and satisfaction levels across different disciplines (such as trainers, psychological service providers, different therapists, i.e., occupational, speech, or physical therapists) and service types (i.e., daycare, rehabilitation, evaluations, and administration).

However, this study is the first on job burnout and job satisfaction among personnel from a center that provides daily services in Iraq’s Kurdistan region. Additionally, searching databases yielded no study related to this topic in the selected area. This preliminary cross-sectional study was conducted in a daycare center between February and May of 2021. This pilot study’s objectives are linked with feasibility and the desire to inform researchers about the best way to conduct a future full-scale project in the area.

### 1.4. The KRI Context

The study was done in Erbil, the capital city of KRI. The data was collected from a private center with a history of eight years of work in the field of DDs, particularly children with ASD. This center has the reputation of being the first daycare center in the city with daily rehabilitation and training programs for children with ASD and some other DDs, such as intellectual disability. Most of the children attend daycare services which are scheduled for four hours a day or for individual clinical sessions. There are other private and semi-governmental centers for children with ASD and different types of DDs. Based on the UN estimation of Iraq’s population of 42 million, 7.1 (17%) million are considered to be the Kurdish population. The Kurdish population is mainly located in the KRI. Erbil’s population is close to 1,200,000, including displaced Kurdish groups such as Yazidis and Syrian Kurdish refugees. Iraq and the KRI as semi-autonomous areas are not exceptional, and there is a shortage of reliable reports on ASD prevalence in both regions. The World Population Review website estimated 89.40 in 10,000 children represent the prevalence rate for this area (Autism Rates by Country, 2022) [34]. Similar to other countries around the world, there is a reported increase in ASD prevalence in the KRI. The Center for Kurdistan Progress [35] estimated that over three thousand children have autism spectrum disorders.

### 1.5. Organizational Challenges and Daycare Management Issues

Service insufficiency and inaccessibility at the national and local levels are apparent challenges in Iraq’s healthcare system [36]. The present difficulties result from many organizational and management factors in the area’s healthcare system [4]. Consequently, the healthcare personnel reacted to these challenges by feeling satisfied with their position in different ways. Their level of burnout, satisfaction with the job, and feeling valued are predicting factors of their ideas about their jobs. There are more reports about the shortage of information in general across the area regarding the different groups’ healthcare needs and inadequate data about healthcare ideas. There are also various healthcare organizational issues, such as a shortage of reasonable programs for managing the predicted healthcare challenges [37].

Gruneberg [38] suggested a conceptual framework based on employees’ feelings about their jobs and on when they attained their systems’ goals. Classical conceptual frameworks mainly consider the mental processes of the employees, utilizing the framework for decision-making and fulfilling their needs. Nevertheless, recent theories are attempting to interpret employees’ feelings toward their job from a different perspective. Judge et al. [39], through the idea of “Social Information Processing,” emphasizes the social context as an essential factor. In this theory, the researchers believe that when the employees are asked about a job opinion, it is formed. They argue that the employees’ perspective regarding the position is influenced by society. Consequently, the job perspective is controlled by external sources, colleagues’ comments, and dominant ideas [40,41,42] that impact employees’ opinions about their jobs.

## 2. Materials and Methods

### 2.1. Participants and Procedure

There were 41 employees registered in a daycare center as the personnel for children with ASD at the time of the study, and 34 (83%) of them who were active personnel at the administration, services, and education/rehabilitation sections participated. The seven other personnel in the center were not active at the time of the study due to different reasons (for instance, three individuals were not active because of maternity leave). All the participants consented to participate and share their ideas about their jobs. Participants working in three departments of General Services (4, 12%), Administration (10, 29%), and Rehabilitation/Training (20, 59%) were asked to fill out scales to understand the level of their job satisfaction and burnout.

The personnel were informed about the aims and objectives of the study, and the participants were provided an oral agreement to participate and also signed written consent. The study was conducted under the Declaration of Helsinki. Without a clear national ethical protocol, we adhere to the seventh revised version WMA of the Helsinki Declaration on Medical Research involving Human Subjects issued on 19 October 2013.

The participants’ age range was 22 to 57, with a mean of 29.6 and a standard deviation of 7.9. Twenty-one members (62%) were female, while 13 (38%) participants were male. Twelve (35%) were married, and 22 (56%) were single or divorced. There were 29 (85%) Iraqi citizens in the sample, while five (15%) of the participants were non-Iraqis (3 Iranian, 1 Turkish, and 1 Syrian). Ten (29%) participants were trained as clinical psychologists, eight of them (23.5%) were trained as special educators, and five members (15%) were physiotherapists and speech and occupational therapists. Seven (21%) participants were trained as administrative and office workers and had taken different training courses in dealing with individuals with ASD. The participants mainly had earned bachelor’s degrees (22, 65%).

### 2.2. Applied Instruments

#### 2.2.1. The Maslach Burnout Inventory-Third Edition (MBI-3)

The MBI-3 [43] consists of 22 items and is designed to evaluate three factors of burnout syndrome: emotional exhaustion (EE), depersonalization/loss of empathy (DP), and personal accomplishment assessment (PA). Based on this logic, the scale is divided into three parts to measure each component of job burnout. The scale manual takes 10 to 15 min to fill out. Since it is recommended to be completed privately, each center member was given the scale in an envelope and it was collected the day after. Based on the scale scoring system, a high score in the EE and DP sections indicates a higher probability of job burnout. In contrast, in the PA part, the scoring is the opposite and a lower score indicates a higher level of job burnout. The reliability for MBI-3 was reported as 0.90 for the Emotional Exhaustion (EE) subscale, 0.79 for the Depersonalization (DP) subscale, and 0.71 for the Personal Accomplishment (PA) subscale. This rate was 0.85 for EE,0.63 for DP, and 0.53 for the PA subscale in the present study.

#### 2.2.2. The Job Descriptive Index (JDI)

This scale is a 90-item inventory designed to measure job satisfaction levels in five parts, each focusing on a dimension of the job. The covered job dimensions are: six various aspects of satisfaction with co-workers (18 items), the work itself (18), work on the present job (18), pay (9), promotion opportunities (9), and supervision (18). The original version developed by Smith et al. [44] attempts to measure “the feelings a worker has about his job” (p. 100), and its most recent revision [45] was utilized. This job satisfaction scale measures the five areas of the job: work, supervision, payment, promotions, and co-workers. Each part includes a checklist of adjectives or adjective phrases, and respondents are requested to check beside each item as follows: “Y” (agreement), “N” (disagreement), and “?” (cannot decide). The Cronbach’s alpha score calculated reliability for the 90 questions in this study and the original scale reported the following: for the work itself, this rate was 0.86 (0.89 original); for co-workers, it was 0.72 (0.89); for present job, it was 0.71 (0.92); for pay, it was 0.57 (0.82); for promotional opportunities, it was 0.52 (0.82); and for supervision, it was 0.75 (0.85).

Both descriptive and inferential statistics (to describe features of data), such as measures of central tendency, and inferential statistics (to infer properties of the reported distribution of probability), such as correlation and Chi-square (Fisher’s Exact Test of Independence was reported because the sample size was small), were used for analysis.

## 3. Results

### 3.1. MBI-3

Table 1 presents the descriptive statistics for the MBI-3 scale.

No statistically significant differences with respect to the level of burnout were seen between the personnel in three different parts of the center (general services, administration, and education/rehabilitation) sections “X2(2, N = 34) 1.735, *p* =0.418”. Hence, younger personnel felt less burnout than their older co-workers “X2(1, N = 34) 4.739, *p* =0.019”. Simultaneously, personnel with more than one responsibility in the center reported a higher level of job burnout (“X2 f(1, N = 34) 6.851, *p =* 0.077”). Personnel at the center generally felt no emotional exhaustion; only 18% (6 members) had this feeling. A similar finding was reported regarding depersonalization/loss of empathy (6 members, 18%). At least 41% (14 members) felt that their position had provided a lower level of personal accomplishment. Table 2 presents the Pearson’s Productive Correlation Coefficiant among subscales of MBI-3 and a statistically significant positive correlation between DP and EE. In contrast, no significant correlation was seen between the two other factors (PA with EE and DP).

A similar statistically significant relationship between the three subscales of MBI-3 was reported using Chi-square. The results are presented in Table 3.

A statistically significant positive relationship between EE and DP indicated that personnel who felt emotionally exhausted had felt depersonalization/loss of empathy. In other words, personnel in all three sections showed that their lowered empathy due to depersonalization was significantly correlated with emotional exhaustion (r = 0.699, n = 34, *p* = 0.000). Correlation is significant at the 0.01 level (2-tailed).

### 3.2. JDI

Table 4 presents the descriptive statistics for the JDI scale.

Considering the overall satisfaction with the job, the personnel generally were satisfied with their jobs, and only items such as promotion (15, 44%) and supervision (12, 35%) indicated higher levels of dissatisfaction.

Table 5 provides data regarding the level of statistically significant correlation among the six subscales of the JDI Pearson’s Productive Correlation Coefficiant among subscales of the scale investigated.

Job in General was one of the scale items that correlated with all the other five factors of the scale (co-workers, present job, payment, promotion, and supervision).

The Chi-square test was used to understand the significant relationship between the subscales of the JDI, and the results are presented in Table 6.

There was a statistically significant relationship between satisfaction with overall work and factors such as co-workers, payment, promotion, and supervision. This indicates that those satisfied with their jobs, in general, were also more satisfied with their co-workers, pay, promotion, and leadership.

Those who were satisfied with their present jobs were more satisfied with their promotion opportunities in their careers.

Younger personnel were more satisfied with their payments compared to the older ones “X2 (1, N = 34) 4.859, *p =* 0.039”.

Higher-educated personnel were less satisfied with their jobs compared to the less-educated group “X2 (1, N = 34) 3.849, *p =* 0.068”.

Personnel with more than one responsibility were less satisfied with their jobs compared to those who had only one responsibility in the center “X2 (1, N = 34) 4.142, *p =* 0.045”.

Female personnel were more dissatisfied with their personal accomplishments compared to the male personnel “X2 (1, N = 34) 4.860, *p =* 0.031”.

Concerning promotion, female personnel were also less satisfied compared to male personnel “X2 (1, N = 34) 3.780, *p =* 0.055”.

No other relationship was found between overall job satisfaction score and factors such as age, gender, and level of education.

A Pearson Correlation Coefficient was computed to assess the linear relationship between the overall score of job burnout and job satisfaction. There was a statistically significant negative correlation between the two variables, r = −0.621, n = 34, *p* < 0.001. Those staff that felt a higher level of satisfaction felt less burnout in their job. Regarding the correlation between the six factors of JDI and three factors of MBI-3, some subtests were significantly positively- or negatively-correlated (Table 7).

## 4. Discussion

The investigation of this study had three aims: to understand the level of job burnout among the healthcare providers in a daycare center for children with ASD, to understand the level of satisfaction of healthcare providers for children with ASD and other DDs in a daycare center, and to determine the contributing factors of job burnout and satisfaction with the jobs in the center.

Although there were degrees of job burnout among staff at the time of the survey, due to the international training opportunities, the hope for updating the system according to the global healthcare system, and some international professionals, the healthcare personnel felt optimistic. Based on the available data, healthcare personnel were more at risk of experiencing job burnout. This finding was congruent with the available literature [19,46]. In comparison, lower levels of emotional exhaustion and depersonalization were reported with a statistically significant correlation between these two factors. This finding indicates that those who feel exhausted also felt less depersonalization and loss of empathy. The reported lower levels of EE and DP among the personnel were different compared to the available data on the healthcare providers [46,47], which might have originated from the wave of optimism in the center at the time of the study.

It was also reported that the personnel at the center felt moderate levels of personal accomplishment in their jobs; the report was supported by findings in other countries [47]. Although no significant negative correlation was found between EE and DP in the present study, it might be concluded that personnel who have reported being less emotionally exhausted and had less feeling of depersonalization were more satisfied with their accomplishments. A further possible survey in this field may focus on boosting the unique sense of accomplishment and its relationship to the levels of EE and DP. Overall, 60% of the participants in this study felt a moderate and higher level of personal accomplishment. This finding might indicate that job burnout might interplay with all three sections in the scale. Therefore, all three subscales needed to be reported separately (as we did in this study) instead of considering an overall score. Those who felt personal accomplishment also had lower senses of feeling emotionally exhausted and detached from co-workers.

Job satisfaction influences job situations, and factors such as work stress negatively impact the pleasure of a job and predict job burnout [48]. Understanding the level of job satisfaction among personnel was the second aim of the current study. Respondents indicated that they are mainly satisfied with their jobs through their choices in the four parts of the scale (co-workers, job in general, present job, and supervision), and the majority of the respondents were satisfied with these aspects of their career. Hence, concerning payment and promotion, the results were different because 50% were happy with the payment, but for the promotion, 44% were less satisfied with the possibility of promotion in their jobs.

Regarding the third aim of the current study, which was to understand the contributing factors of job burnout and satisfaction with the job, findings indicated that job satisfaction was reported to have a statistically significant relationship with factors such as payment, supervision, and co-workers. These findings were incongruent with previous data regarding satisfaction with the job. Payment in other studies is also considered a significant predictor of job satisfaction [49,50,51]; supervision [52,53,54] and co-workers [55,56] were also reported to be significant predictors of job satisfaction.

It is believed that a variety of factors contribute to job burnout. Factors such as the lack of job resources as well as a lack of control [57], pressure and lack of support [58], work and life balance [59], and dysfunctional workplace dynamics [60] may all contribute to burnout.

It is reported that the inability to influence decisions that affect one’s job, such as work schedule, assignments, or workload, might contribute to job burnout. Similar findings reported a lack of needed resources for job performance.

The ambiguity of job expectations, such as uncertainty about the sources and levels of authority or fluctuations in supervisors’ expectations, increases job burnout. Significant consequences are reported for job burnout, such as lack of job satisfaction, increased stress, and psychosomatic reactions such as insomnia, sadness and anger, and substance abuse [61].

Evaluating personnel options and understanding the expectations and a call for reaching compromises or solutions, along with the availability of support from supervisors, co-workers, and collaboration, might help personnel to cope with job burnout [60].

Reviewing working conditions and providing different opportunities for hearing from the personnel might provide ideas for having a positive impact on job burnout or boosting satisfaction with the job. Eventually, providing learning opportunities for the personnel may help make a positive impact on their satisfaction and burnout. Hence, as Samarakoon [62] indicated, micromanaging style as the core of the classical management approach might contribute to the reported findings. In this style, an individual, based on their position as the founder or financial source, has total authority, decision-making, and absolute control over the subordinates in the organization. This was the managing style in the center and might be considered a contributing factor to the personnel’s negative feelings about the job and reports of job burnout.

### Limitations

These findings need to be considered concerning their limitations. A small sample size impacted the findings’ generalized application. The sample was from one center and revealed the convenient sample’s data. Studies with bigger sample sizes with different groups of service providers in various cities across the region might be able to yield more robust results. Since the sample of the present study was from the private sector, studying the governmental healthcare providers might present different findings. The survey scales were not tested in the Kurdish culture, and available norms of the scales were considered. Previous data indicated that JDI was applied more to business organizations, but this study used it in the healthcare sector. Hence, the present study was novel and groundbreaking because healthcare providers in the area have rarely been under investigation to understand their impressions about their jobs. This study might be helpful in highlighting the importance of understanding the healthcare providers’ perspectives about their careers in the area and be considered as a starting point for further investigations.

## 5. Conclusions

Low satisfaction with healthcare providers among parents and caregivers of individuals with ASD has been recognized as contributing to caregiving stress and challenges [63]. Job burnout and lack of job satisfaction have been considered the critical elements of job morale among healthcare providers, and these elements determine healthcare providers’ level of performance. In conclusion, healthcare centers and managers of these services need to understand the costs of personnel burnout and their lower levels of satisfied with job imposed on the healthcare system. Psychological factors such as job burnout and job satisfaction need to be understood. These studies are crucial, especially in countries with limited sources, and a professional understanding of these factors is more critical. Understanding the impeding factors of job satisfaction, contributing factors to job burnout, and the process and associated risk factors at its early stages are essential to allocating sufficient internal and external support to address recognized factors. Positive motivational factors, such as providing promotion opportunities, establishing a solid internal network, and income improvement, can protect the healthcare system and improve job satisfaction. Providing training opportunities for healthcare personnel groups, teaching adaptive coping, and enhancing resilience are proper strategies in healthcare policies.

Further studies are needed to understand the predicting factors of job burnout and satisfaction with healthcare duties. Studies using different methodological approaches might be able to yield more information.

## Figures and Tables

**Table 1 brainsci-13-00251-t001:** Descriptive statistics of The Maslach Burnout Inventory-Third Edition (MBI-3) Scale N = 34.

Subscale	Mean	SD	Max	Min	Low Level	Moderate Level	High Level
Emotional Exhaustion (EE)	17.32	11	41	1	19 (56%)	9 (26.5%)	6 (18%)
Depersonalization/loss of empathy (DP)	5.79	5.20	15	0	17 (50%)	11 (32%)	6 (18%)
Personal Accomplishment (PA)	34.88	6.39	46	18	14 (40%)	10 (29%)	10 (29%)

**Table 2 brainsci-13-00251-t002:** Pearson’s Productive Correlation Coefficiant between subscales of the MBI scale.

	Depersonalization/Loss of Empathy (DP)	Personal Accomplishment (PA)
Emotional Exhaustion (EE) and	0.699 **	−0.255
Depersonalization/loss of empathy (DP)		−0.294

** Correlation is significant at the 0.01 level (2-tailed).

**Table 3 brainsci-13-00251-t003:** Chi-square results and level of the significant relationship between the subscales of MBI-3.

Subscale	X2	Significance
Emotional Exhaustion (EE) and Depersonalization/Loss of empathy (DP)	34.18	0.000
Emotional Exhaustion (EE) and Personal Accomplishment (PA)	5.72	0.221
Depersonalization/Loss of empathy (DP) and Personal Accomplishment (PA)	5.91	0.205

**Table 4 brainsci-13-00251-t004:** Descriptive statistics of the Job Description Index (JDI) N = 34.

Subscale	Mean	SD	Max	Min	Low Level of Satisfaction	Moderate Level of Satisfaction	High Level of Satisfaction
Co-workers	29.47	5.13	42	23	11 (32%)	10 (30%)	13 (38%)
Job in General	29.35	4.22	39	23	11 (32%)	10 (30%)	13 (38%)
Present job	27.94	4.74	43	22	9 (26.5%)	9 (26.5%)	16 (47%)
Payment	14.85	2.66	23	9	7 (21%)	17 (50%)	10 (29%)
Promotion	14.38	4.46	22	10	15 (44%)	8 (23.5%)	11 (32%)
Supervision	30.32	6.31	54	22	12 (35%)	6 (18%)	16 (47%)

**Table 5 brainsci-13-00251-t005:** Pearson’s Productive Correlation Coefficiant between subscales of the JDI scale.

	Job in General	Present Job	Payment	Promotion	Supervision
Co-workers	0.539 **	0.236	0.341 *	0.212	0.537 **
Job in General		0.573 **	0.341 *	0.455 **	0.732 **
Present job			0.321	0.645 **	0.321
Payment				0.549 **	0.310
Promotion					0.357 *

** Correlation is significant at the 0.01 level (2-tailed). * Correlation is significant at the 0.05 level (2-tailed).

**Table 6 brainsci-13-00251-t006:** Chi-square results and level of the significant relationship between the subscales of JDI.

Subscale	X2	Significance
Job in General and Co-workers	11.73	0.004
Job in General and Payment	9.78	0.095
Job in General and Promotion	11.53	0.021
Job in General and Supervision	20.90	0.000
Present job and Promotion	21.87	0.000

**Table 7 brainsci-13-00251-t007:** Pearson’s Productive Correlation Coefficient between the sub-scales of the factors of the Job Descriptive Index (JDI)and the Maslach Burnout Inventory-Third Edition (MBI-3).

	Emotional Exhaustion (EE)	Depersonalization/Loss of Empathy (DP)	Personal Accomplishment Assessment (PA)
satisfaction with co-workers	0.032 0.856	−0.061 0.734	0.014 0.936
the work itself	0.059 0.739	−0.059 0.741	−0.118 0.507
work on the present job	−0.404 * 0.018	−0.415 * 0.015	−0.349 * 0.043
pay	−0.126 0.479	−0.118 0.507	0.181 0.307
promotion opportunities	−0.349 * 0.043	−0.417 * 0.014	0.398 * 0.020
supervision	−0.230 0.190	−0.424 * 0.013	0.316 * 0.012

* Correlation is significant at the 0.05 level (2-tailed).

## Data Availability

Data is unavailable due to privacy or ethical restrictions regarding the personnel ideas on the system. The center name and data remained confidential.

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
