# Peer review of "Job Burnout and Job Satisfaction among Healthcare Service Providers in a Daycare Center for Individuals with Autism Spectrum Disorders in Low-Resource Settings"

_brainsci, 2023, doi:10.3390/brainsci13020251_

Round 1

Reviewer 1 Report

Study Title:  Job Burnout and Job Satisfaction Among Healthcare Service Providers in a Daycare Center for Individuals with Autism Spectrum Disorders in the Low-Resource Settings

General

1.       In the current study, the author correlated the factors within the instruments (i.e., EE, DP, and PA within MBI-3 and Work, supervision, payment, promotions, and co-workers within JDI). It would be interesting if the authors correlate the factors across instruments to understand how different job descriptive indexes (JDI) might affect the burnout level (MBI-3).

Abstract

1.       I suggest the abstract be reorganized to contain important information such as subject information, names of the assessment tool, statistics, and quantitative results.

Introduction

1.       Please clearly describe the research questions and provide the hypothesis for each research question.

2.       Please make sure all abbreviation is explained, for example, DD in line 52 is not mentioned before.

Materials and Methods

1.       I suggest moving the first part of the method section (Lines 171 to 181) to the introduction section as it is not relevant to the current study design.

2.       Did the author only include full-time personnel? Please explain the rationale for the decision.

3.       The # of participants working in 3 departments (3 in general services, 9 in administration, and 20 in Rehabilitation/Training) does not add to the total # of participants stated by the author (n=34).

4.       Please include detailed information of statistical analyses in the method section.

Results

1.       I suggest moving Lines 229 to 232 and Lines 255 and 259 to the method section, as it is not about the current findings.

2.       On lines 236-237, the author stated, “No statistically significant differences were seen between the personnel in different parts of the center (general services, administration, and education/rehabilitation) sections.” It is not clear to me what stats the authors used to take the center into account (please mention this in the method section). Please also provide statistics and quantitative results associated with this statement (e.g., mean, SD, and p-value).

3.       Please specify the direction of the correlation, ex: subjects who had greater emotional exhaustion had greater loss of empathy (Lines 241-244).

4.       The abbreviation for Maslach Burnout Inventory switched between “MBI-3” and “BMI-3” in the result section. Please make sure they are consistent.

5.       For Table 2, please provide a title.

6.       For Table 3, the first and second Chi-square listed are between the same set of subtests (EE and DP), but the Stats are very different. Please fix the table presentation.

7.       Similarly, in lines 285-286, the author stated, “No other relationship was found between job satisfaction and factors such as age, gender, and level of education.” It is not clear to me what stats the authors used to take the age, gender, and level of education into account (please mention this in the method section). Please also provide statistics and quantitative results associated with this statement (e.g., mean, SD, and p-value). 

Author Response

Thank you for the helpful comments.  We have done our best to address them in the following table.

Reviewer No:1

Response to the comment/suggestion

General

1.       In the current study, the author correlated the factors within the instruments (i.e., EE, DP, and PA within MBI-3 and Work, supervision, payment, promotions, and co-workers within JDI).  It would be interesting if the authors correlated the factors across instruments to understand how different job descriptive indexes (JDI) might affect the burnout level (MBI-3).

A new analysis was added to the end of the results section.

 A Pearson correlation coefficient was computed to assess the linear relationship between the overall score of job burnout and job satisfaction.  There was a statistically significant negative correlation between the two variables, r=-0.621, n=34, p = 0.00.  Those staff with higher satisfaction levels felt less burnout in their job.  Regarding the correlation between the six factors of JDI and three factors of MBI-3, a new table was added.

emotional exhaustion (EE)

depersonalization/loss of empathy (DP)

personal accomplishment assessment (PA)

satisfaction with co-workers

.032

.856

-.061

.734

.014

.936

the work itself

.059

.739

-.059

.741

-.118

.507

work on the present job

-.404*

.018

-.415*

.015

-.349*

.043

pay

-.126

.479

-.118

.507

.181

.307

promotion opportunities

-.349*

.043

-.417*

.014

.398*

.020

supervision

-.230

.190

-.424*

.013

.316*

.012

Correlation is significant at the 0.05 level (2-tailed).                        

Abstract

1.       I suggest the abstract be reorganized to contain important information such as subject information, names of the assessment tool, statistics, and quantitative results.

The abstract was reorganized based on the requested information.  Subject information was added, scales were mentioned, and more statistical information about the findings was added.

Introduction

1.       Please clearly describe the research questions and provide the hypothesis for each research question.

The objectives and research question were clarified and rephrased in the form of research questions:

·        There is a significant difference between the level of job burnout among rehabilitation and training staff of healthcare providers compared to the services and administration staff.

•             There is a significant difference between the level of job satisfaction among rehabilitation and training staff of healthcare providers compared to the services and administration staff.

•             There are significant correlations between factors of job burnout and satisfaction with the job

2.       Please make sure all abbreviation is explained; for example, DD in line 52 is not mentioned before.

The term developmental disabilities was introduced before DD in line 52.

Materials and Methods

1.       I suggest moving the first part of the method section (Lines 171 to 181) to the introduction section as it is not relevant to the current study design.

The first paragraph moved to the end of the introduction section under the Organizational challenges and Daycare management sub-section.

2.       Did the author only include full-time personnel?  Please explain the rationale for the decision.

The full-time was deleted since the working period was not considered.

There was only 2 part-time personnel that were also participated in the study.  The 7 other staff were not active at the time of the study in the center for different reasons (3 because of maternity leave).

3.       The # of participants working in 3 departments (3 in general services, 9 in administration, and 20 in Rehabilitation/Training) does not add to the total # of participants stated by the author (n=34).

The mistake regarding the personnel was corrected.  Thirty-four participants were working in 3 departments of General Services (4, 12%), administration (10, 29%), and Rehabilitation/Training (20, 59%).

4.       Please include detailed information of statistical analyses in the method section.

Detailed analysis was added to the method section.

Both descriptive and inferential statistics (to describe features of data), such as measures of central tendency, and inferential statistics (to infer properties of the reported distribution of probability), such as correlation and Chi-square (Fisher’s Exact Test of Independence reported because the sample size was small), were used for analysis.

Results

1.       I suggest moving Lines 229 to 232 and Lines 255 and 259 to the method section, as it is not about the current findings.

The suggested move was considered

2.       On lines 236-237, the author stated, “No statistically significant differences were seen between the personnel in different parts of the center (general services, administration, and education/rehabilitation) sections.” It is not clear to me what stats the authors used to take the center into account (please mention this in the method section).  Please also provide statistics and quantitative results associated with this statement (e.g., mean, SD, and p-value).

The Chi-square between the three groups was used to understand the possible significant defenses.  This part was updated as follows:

No statistically significant differences concerning the level of burnout were seen between the personnel in 3 different parts of the center (general services, administration, and education/rehabilitation) sections (“X2= df=2, N = 34, 1.735, p = .  418”).

3.       Please specify the direction of the correlation, ex: subjects who had greater emotional exhaustion had greater loss of empathy (Lines 241-244).

We have some more analysis to respond to your comment.  With respect to your comment, the following information was added:

A statistically significant positive relationship between EE and DP indicated that personnel who felt emotionally exhausted had felt depersonalization/loss of empathy.  In other words, personnel in all three sections showed that their lowered empathy due to depersonalization was significantly correlated with emotional exhaustion (r=,0.699, n=34, p=.000).  Correlation is significant at the 0.01 level (2-tailed). 

The following information was also added in this section:

Hence, younger personnel felt less burnout compared to their older co-workers “X2=df=1, N = 34, 4.739, p =.  019”.  Simultaneously, personnel with more than one duty in the center reported a higher level of job burnout “X2 df=1, N = 34, 6.851, p =.  077”.

4.       The abbreviation for Maslach Burnout Inventory switched between “MBI-3” and “BMI-3” in the result section.  Please make sure they are consistent.

The abbreviation was corrected

5.       For Table 2, please provide a title.

6.       For Table 3, the first and second Chi-square listed are between the same set of subtests (EE and DP), but the Stats are very different.  Please fix the table presentation.

5.  The following title was considered for table 2:

The Person’s productive value among 3 subscales of MBI

6.The table was fixed, and the Chi-square between Emotional Exhaustion (EE) and Personal Accomplishment (PA) was reported.

7.       Similarly, in lines 285-286, the author stated, “No other relationship was found between job satisfaction and factors such as age, gender, and level of education.” It is not clear to me what stats the authors used to take the age, gender, and level of education into account (please mention this in the method section).  Please also provide statistics and quantitative results associated with this statement (e.g., mean, SD, and p-value).

The method section was updated based on your comment, and more analysis was added to the job satisfaction part.

To make the final statement of the result section with was about the overall job satisfaction score following analysis was done considering the participants' demographic factors and sub-scales of JDI:

Younger staff were more satisfied with their payment compared to the older ones “X2=df=1, N = 34, 4.859, p =.  039”

Higher educated staff were less satisfied with their job compared to the less educated group “X2=df=1, N = 34, 3.849, p =.  068”

Staff with more than one duty were less satisfied with their job compared to those who had only one responsibility in the center “X2= (df=1, N = 34) = 4.142, p =.  045”

Female staff were more dissatisfied with their personal accomplishments compared to the male staff “X2=df=1, N = 34, 4.860, p =.  031”

With respect to the promotion, female staff were also less satisfied compared to male staff “X2=df=1, N = 34, 3.780, p =.  055”

No other relationship was found between overall job satisfaction score and factors such as age, gender, and level of education.

Reviewer 2 Report

The subject of the article refers to an interesting and important issue, which is burnout and job satisfaction among healthcare service providers in a daycare center for patients with autism spectrum disorders (ASD). This is due, among others, to the fact that the number of people with ASD is dynamically growing in different countries, regardless of their level of socio-economic development. Therefore, a closer understanding of the factors determining job satisfaction among employees of such centers is of great cognitive and practical importance. However, the reviewed article has some deficiencies that reduce its scientific value. First of all, it refers to its formal structure. For example, the paragraph on page 4 (lines 171-181) should be placed in the theoretical part (Introduction) instead of "Materials and methods". Also, the paragraph on page 5 (lines 218-227) should be moved to the "Participants" section instead of "Results". On page 2 (line 42) the authors introduce the abbreviation HICs, the meaning of which was not previously explained. The same applies to the abbreviation DD (page 2, lines 52 and 54). Although the authors know the meaning of these abbreviations, they should bear in mind readers who may not understand them. The authors also use the abbreviation DS for standard deviation (Table 1 and 4), while in the scientific literature this abbreviation is denoted as SD. Other formal deficiencies include the lack of the title in Table 2. I disagree with the authors' statement that job burnout and job satisfaction belong to organizational factors (page 10, line 379). In fact, they belong to psychological (personal) factors. In the "Limitations" section, the authors rightly state that a relatively small sample size limits the value of the results obtained. On this basis, in the methodological section, they should define their research as a pilot study. 

Author Response

Thank you for the helpful comments.  We have done our best to address them in the following table.

The subject of the article refers to an interesting and important issue, which is burnout and job satisfaction among healthcare service providers in a daycare center for patients with autism spectrum disorders (ASD). This is due, among others, to the fact that the number of people with ASD is dynamically growing in different countries, regardless of their level of socio-economic development.

Therefore, a closer understanding of the factors determining job satisfaction among employees of such centers is of great cognitive and practical importance.

However, the reviewed article has some deficiencies that reduce its scientific value.

First of all, it refers to its formal structure.  For example, the paragraph on page 4 (lines 171-181) should be placed in the theoretical part (Introduction) instead of "Materials and methods".

The first paragraph, page 4 (lines 171-181)  moved to the end of the introduction section under the Organizational challenges and Daycare management sub-section.

Also, the paragraph on page 5 (lines 218-227) should be moved to the "Participants" section instead of "Results".

Participants' analysis results moved to the end of the Participants section.

On page 2 (line 42) the authors introduce the abbreviation HICs, the meaning of which was not previously explained.

The same applies to the abbreviation DD (page 2, lines 52 and 54).  Although the authors know the meaning of these abbreviations, they should bear in mind readers who may not understand them.

Low- and middle-income countries (LMICs) and high-income countries (HICs) were introduced in full before presenting the abbreviations.

The term developmental disabilities was introduced before DD in line 52.

The authors also use the abbreviation DS for standard deviation (Tables 1 and 4), while in the scientific literature, this abbreviation is denoted as SD.

Other formal deficiencies include the lack of the title in Table 2.

SD substituted with DS in both Tables 1 and 4.

The following title was considered for table 2:

The Person’s productive value among 3 subscales of MBI

I disagree with the authors' statement that job burnout and job satisfaction belong to organizational factors (page 10, line 379).  In fact, they belong to psychological (personal) factors.

The sentence was updated as the following:

Psychological factors such as job burnout and job satisfaction need to be understood.

In the "Limitations" section, the authors rightly state that a relatively small sample size limits the value of the results obtained.  On this basis, in the methodological section, they should define their research as a pilot study

:

The final part of the “present study” section was updated:

 This is a preliminary cross-sectional study conducted in a daycare center between February and May of 2021.  This pilot has its objectives linked with feasibility and the desire to inform researchers about the best way to conduct the future, full-scale project in the area.

Author Response

Thank you for the helpful comments.  We have done our best to address them in the following table.

Check Repetition:

These factors have received scant attention in the literature and 16

particularly in the healthcare systems of the Kurdistan Region of Iraq (KRI) as one low- or middle-

income country (LMIC).

The abstract was reorganized, and the appointed sentences changed to:

Abstract: Job satisfaction and burnout are components of job morale. These factors, in general, and among healthcare provider personnel, are psychological factors of the job and under the influence of different conditions and the organizational management of the healthcare systems. Both job burnout and job satisfaction among healthcare service providers have received scant attention in the literature, particularly in the healthcare systems of the Kurdistan Region of Iraq (KRI) as one low- or middle-income country (LMIC).

Why do the authors begin with disgnosis for ASD criteria?  if the focus of interest are providers of healthservice.  Shouldn't be better to begin focusing on the sample of interest?

We decided to start with the diagnosis to stress the importance of the study with respect to the increasing prevalence rate globally and its core symptoms (particularly behavioral problems) that are considered to be the root of the associated challenges of caregivers for both parents and professional groups.

This is the focus of interest, maybe to hierarchichaly give more weight to this sentence:

There 43 are different aspects of the impacts of this population on various groups, such as care-44 givers, service providers, and other factors that deserve to be understood.

The weight of the sentence was increased, and it is expanded as follows:

There are different aspects of the impacts of this population on various groups, such as caregivers, service providers, and other factors that deserve to be understood, and the wealth of data on the family, friends, and caregivers have shown that they are significantly and negatively impacted by caregiving.  

DD requires previous explanation at this point.

The term developmental disabilities was introduced before DD in line 52.

Check for imnprove sentences, which are very short and require referencing.

The appointed sentence was made longer and merged with its following sentence to make it closer to its reference (number 15:

Sabitova, A., Hickling, L.M. & Priebe, S. Job morale: a scoping review of how the concept developed and is used in healthcare research. BMC Public Health 20, 1166 (2020).  https://doi.org/10.1186/s12889-020-09256-6

The objective of the study is not clearly stated.

There are also various 167 healthcare organizational issues, such as a shortage of reasonable programs for manag-168 ing the predicted healthcare challenges [37].

The objectives and research question were clarified and rephrased in the form of research questions:

·        There is a significant difference between the level of job burnout among rehabilitation and training staff of healthcare providers compared to the services and administration staff.

•             There is a significant difference between the level of job satisfaction among rehabilitation and training staff of healthcare providers compared to the services and administration staff.

•             There are significant correlations between factors of job burnout and satisfaction with the job

This information looks to deliver conceptualization information, shouldn't be better to present this information in the introduction?

The first paragraph in which theoretical frameworks are presented, moved to the end of the introduction section under the Organizational challenges and Daycare management sub-section.

What is the research design?

We made it clear in the final part of the “present study” section research design:

This preliminary cross-sectional study was conducted in a daycare center between February and May of 2021. This pilot has its objectives linked with feasibility and the desire to inform researchers about the best way to conduct the future, full-scale project in the area.

Human participants require IRB approval, was this protocol approved?  authors should provide evidence for that.

Unfortunately, the answer to this question is negative.  We made it clear that the study is done in an area with a lack of an ethical regulation system and in the absence of such a regulation, we tried to consider the available international research ethics standards.  Instead of the American IRB system, we translated and considered the revised version of WMA.  The Kurdish translation (in which the rights of the participants and obligations of the researchers are mentioned) and the signed consent forms s are available and will be shared as an approval based on the formal request of the journal and reviewer.

Round 2

Reviewer 1 Report

The authors did a good job answering the questions/making modifications. I believe that the current version is much stronger.

Reviewer 2 Report

After reading the revised version of the article in detail, I noticed that the authors had made the corrections and changes which I had proposed. They rightly emphasized that for methodological reasons, the conducted research is of a pilot study. It is good that in the final part they postulate conducting research on a larger population of respondents, also in social health care facilities. In this context, after the correction of the text, the scientific value of the article is higher. On this basis, I support the publication of the article in its current version.

Reviewer 3 Report

I appreciate the effort made by the authors in improving the article. The authors have fully addressed the details that I identified in the first version of the manuscript, my main concerns regarding research questions and objectives are now clearly stated, and ethical considerations are attended as well.

I could only address some comments looking for another opportuniity to improve this interesting research report:

1. Authors listed research question using vignettes format, I would suggest stating their questions using prose format (lines 94-101)

2. Some minor corrections using statistical abbreviations (lines 296-299) for Chi Sq symbols, and spaces (according Jounal´s style), lines 306, 332, 377 Pearson instead of Pierson, 331 Pearson instead of Person.

3. Lines 354-364, please join sentences together as as a conituous report.

4. Line 374, please report p < .001 (p .000 is not a possible value).

5. Please format table 7, first column is not understood, it is difficult to identify what variables are being correlated.  Add horizontals as corresponding before table note.